



# Testing the reconstruction of modelled particulate organic carbon from surface ecosystem components using PlankTOM12 and Machine Learning

Anna Denvil-Sommer[1], Erik T. Buitenhuis[1], Rainer Kiko[2,3], Fabien Lombard[2,4], Lionel Guidi[2], Corinne Le Quéré[1]

[1]School of Environmental Science, University of East Anglia, Norwich, UK
[2]Sorbonne Université, Centre National de la Recherche Scientifique (CNRS), Laboratoire d'Océanographie de Villefranche (LOV), Villefranche-sur-Mer, France
[3]GEOMAR Helmholtz Center for Ocean Research, Kiel, Germany
[4]Institut Universitaire de France (IUF), Paris, France

*Correspondence to*: Anna Denvil-Sommer (anna.sommer.lab@gmail.com)

**Abstract.** Understanding the relationship between surface marine ecosystems and the export of carbon to depth by sinking organic particles is key to represent the effect of ecosystem dynamics and diversity, and their evolution under multiple stressors, on the carbon cycle and climate in models. Recent observational technologies have greatly increased the amount of data available, both for the abundance of diverse plankton groups and for the concentration and properties of particulate organic carbon in the ocean interior. Here we use synthetic model data to test the potential of using Machine Learning (ML) to reproduce concentrations of particulate organic carbon within the ocean interior based on surface ecosystem and environmental data. We test two machine learning methods that differ in their approaches to data-fitting, the Random Forest and XGBoost methods. The synthetic data is sampled from the PlankTOM12 global biogeochemical model using the time and coordinates of existing observations. We test 27 different combinations of possible drivers to reconstruct small ($POC_S$) and large ($POC_L$) particulate organic carbon concentrations. We show that ML can successfully be used to reproduce modelled particulate organic carbon over most of the ocean based on ecosystem and modelled environmental drivers. XGBoost showed better results compared to Random Forest thanks to its gradient boosting trees architecture. The inclusion of Plankton Functional Types (PFTs) in driver sets improved the accuracy of the model reconstruction by 58% on average for $POC_S$, and by 22% for $POC_L$. Results were less robust over the Equatorial Pacific and some parts of the high latitudes. For $POC_S$ reconstruction, the most important drivers were the depth level, temperature, microzooplankton and $PO_4$, while for $POC_L$ it was the depth level, temperature, mixed-layer depth, microzooplankton, phaeocystis, $PO_4$ and chlorophyll *a* averaged over the mixed-layer depth. These results suggest that it will be possible to identify linkages between surface environmental and ecosystem structure and particulate organic carbon distribution within the ocean interior using real observations, and to use this knowledge to improve both our understanding of ecosystem dynamics and of their functional representation within models.

## 1. Introduction.

Progress in numerical ocean modelling over multiple decades coupled with fundamental knowledge of fluid dynamics have led to an explicit representation of ocean dynamics in Earth System Models and of most of its key features, apart from small-scale features which are parametrized. In contrast, ecosystem dynamics in ocean biogeochemical models are much more reliant on empirical data for growth and loss processes, with the theoretical basis limited to the dynamic representation of interactions among lower trophic levels (zooplankton and smaller organisms) and their influence on carbon pools and fluxes (Le Quéré et al., 2005; Hood et al., 2006). The recent advances in observational technologies including imaging data (Guidi et al., 2016), genomics (Kirchman et al., 2016), and field study (Mutshinda et al., 2017; Batten et al., 2019, Lombard et al., 2019), offer new opportunities to improve our understanding of marine ecosystem dynamics, and to better represent its influence on carbon pools and fluxes in models that are used to project future climate change and associated impacts on ecosystems.

One strategy to represent lower trophic interactions in global biogeochemical models is to combine different species into Plankton Functional Types (PFTs) based on their unique influence on global biogeochemical cycles (Le Quéré et al., 2005; Hood et al., 2006). This approach enables the representation of plankton types that are unique, have an





influence on other PFTs within the ecosystem and are of quantitative importance for carbon flux and other biogeochemical fluxes. The PlankTOM12 model is among the most detailed in this category of models with its inclusion of an explicit representation of twelve PFTs: six phytoplankton, five zooplankton, and bacteria. PlankTOM12 builds on the published version PlankTOM10 (Le Quéré et al., 2016) that has been extended to include gelatinous zooplankton (Wright et al., 2021) and pteropods (Buitenhuis et al., 2019). Much effort has been put into the development of PFTs and associated representation of surface ecosystem dynamics, which has led to the demonstration that: (1) the representation of trophic levels was a key determinant of the low chlorophyll *a* concentration observed in the Southern Ocean summer (Le Quéré et al., 2016); (2) $CaCO_3$ dissolution above the lysocline is needed to reproduce observations of both biomass and export of PFT calcifiers, and (3) gelatinous zooplankton plays an important role in determining surface biomass of other PFTs (Wright et al., 2021).

In contrast, the transfer of organic matter resulting from surface ecosystem dynamics into carbon exported to the deep ocean via the sinking of particulate organic matter has received much less attention, so that improvements in the representation of the PFTs do not necessarily translate into improvements in sinking of particulate matter (Wright et al., 2021). The export flux of particulate organic carbon from the surface ocean to depth is around 10 PgC yr$^{-1}$ (Schlitzer, 2002), which is as large as the $CO_2$ emitted to the atmosphere by human activities and nearly four times larger than the mean oceanic $CO_2$ sink in recent decades (Friedlingstein et al., 2022). Changes in carbon exported to depth can have a large impact on air-sea $CO_2$ fluxes and on the amount of $CO_2$ emissions that remain in the atmosphere where they cause climate change.

The growing amount of observations provides the opportunity to develop a new approach to explore the linkages between surface ecosystem dynamics and the distribution of particulate organic carbon in the ocean, and to improve the representation of particle sinking fluxes in models. However, there is a risk of over-interpreting the data by applying Machine Learning (ML) methods directly to link the observed surface environment and ecosystem structure with the observed particulate organic carbon distribution. The use of synthetic observations based on model data therefore provides a minimum test to assess the likely success and usefulness of such an approach.

ML has been widely used in biogeochemical and geophysical applications and provided efficient results in reconstructions of ocean surface pCO₂ (Friedrich and Oschlies, 2009; Telszewski et al., 2009; Landschützer et al., 2013; Denvil-Sommer et al., 2019) and of particulate organic carbon (Sauzède et al., 2016, 2017) as well as in the analysis of driver importance (Sauzède et al., 2020).

Here we use model data to verify the hypothesis that the composition of surface ecosystems and environmental conditions are indeed reflected in the abundance and size of the organic particles in the ocean interior. We reconstruct the concentration of organic particles as represented by small (POC$_S$, particles < 256μm) and large (POC$_L$, particles > 256μm) particulate carbon in the PlankTOM12 model. Using this information alongside with modelled environmental and ecosystem conditions we develop a ML method to reproduce POC$_S$ and POC$_L$ over the global ocean and verify the hypothesis. This constitutes a necessary although not sufficient test that the approach can subsequently be used to reveal linkages using real observations and to inform model developments.

## 2. Data and Methods.

In this section we describe a set of variables that will be used to test the ML method's ability to reconstruct particulate organic carbon concentrations based on ocean model data. We create a set of synthetic data by sampling a model at the time and location of real-world observations. We discuss the availability and distribution of real-world observations and their limitations. In this section we also describe the PlankTOM12 global ocean biogeochemical model and how we use it to develop a ML method and test its ability to reconstruct small and large particulate organic carbon with a limited number of observations. To provide resemblance to the real data availability we focus on the period 2009-2013 which guarantees additional sampling of co-located biological, chemical, and environmental variables from the Tara expeditions (Sunagawa et al., 2020).

Two sets of data are needed to test the Machine Learning method: a set of targets and a set of drivers. The drivers represent the input variables to the ML method (here the biological, chemical, and environmental variables). The targets represent the variables we are trying to reconstruct (here the particulate organic matter POC$_S$ and POC$_L$). The ML will then determine the relationship between the drivers and targets, which can then be applied in regions where drivers are available to infer targets where the later data do not exist.





### 2.1.1. Measurements of particle size distributions and concentrations (the targets).

We use observations of particle distribution in two ways. First to determine the time and location of the observations, and second to verify that the ocean model is of sufficient quality to be used in this analysis. The sampling of the particulate organic carbon concentration is based on the data from an Underwater Vision Profiler 5 (UVP5) (Gorsky et al., 2000, 1992; Picheral et al., 2010; Kiko et al., 2022). UVP5 measures particles of size from 50 μm to a few mm. For the purpose of comparing the UVP5 data with the PlankTOM12 model data, we converted measured biovolume concentration (mm$^3$/L) of particles to carbon biomass concentrations (μmol/L) using the empirical equation from Alldredge (1998) for particulate organic carbon:

$$BM\ (\mu g) = 0.99*BV\ (mm^3)^{0.52}$$

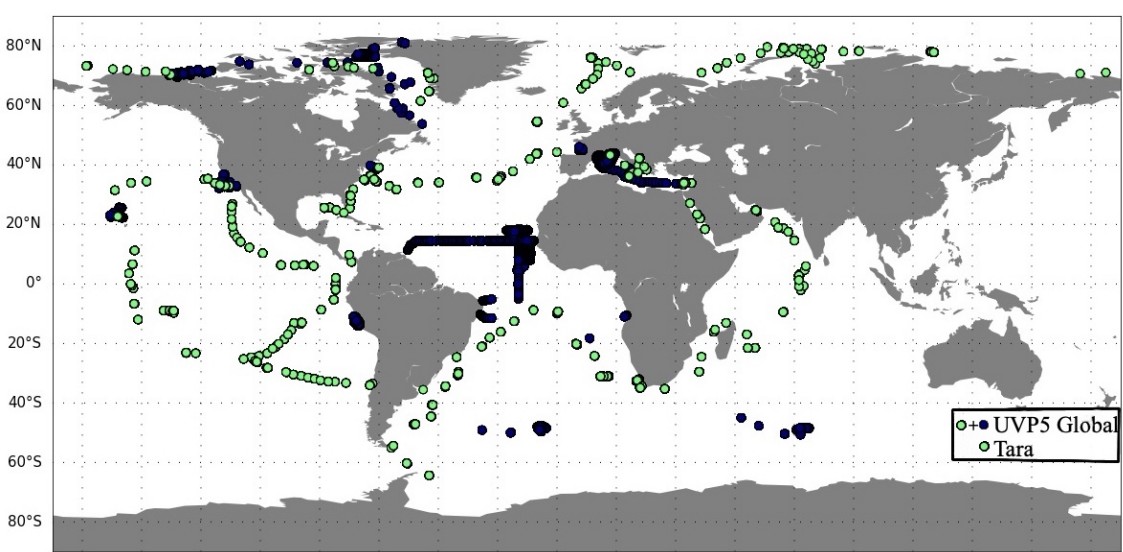

**Figure 1. Location of the observations from the UVP5 database over the period 2009-2013. Green dots correspond to Tara expeditions, and were included in the global UVP5 database.**

We summed size classes from 50.8 μm to 256 μm for the small particulate organic carbon (POC$_S$) and from 256 μm to 5.16 mm for large particulate organic carbon (POC$_L$). POCs below 100 μm are not well captured by the UVP sensor, which therefore underestimates this size-class of aggregated particles. We extrapolated the total size of particles up to 0.001mm by using the size spectra theory to provide a better estimate of POC biomass concentration in line with the model. Following Guidi et al. (2008) we used the abundance of particles sized from 0.250 to 1.5 mm excluding rare particles to estimate the coefficients of logarithmic relationship between the size of particles and its abundance:

$$log(abundance) \ = \ a * log(size) \ + \ b$$

Using this equation we estimated the abundance of particles of size less than 100 μm.

There are 2603 vertical profiles of UVP5 measurements during 2009-2013, including 752 profiles which are co-located with the stations from the Tara expeditions that provide the environmental and ecosystem variables (Figure 1; Section 2.1.2). The measurements are sparse in time and space. There are no measurements in the Southern Ocean, Western Pacific Ocean and Eastern Indian Oceans.

### 2.1.2. Measurements of environmental and ecosystem variables (the drivers).

We use observations of environmental and ecosystem variables to determine the time and location of the observations that are colocated with the target variables. To represent the main physical and chemical drivers responsible for the concentration and variability of POC$_S$ and POC$_L$ we use measurements of ocean temperature, chlorophyll $a$, phosphate PO$_4$, nitrates NO$_3$, mixed-layer depth (MLD). These variables were measured during Tara expeditions along with the



particle size distributions and concentrations using UVP instruments onboard these cruises. However, chlorophyll *a*,
PO$_4$, and nitrates were not measured systematically at each depth level. Thus, their averages over MLD are tested as
possible drivers as well. To represent the biological drivers, we use information on PFTs.
### 2.1.3. The NEMO-PlankTOM12 Global biogeochemical model.
We used the output from the NEMO-PlankTOM12 coupled physical–biogeochemical model of the global ocean at
daily and monthly time resolution. NEMO represents physical transport processes and is used in its v3.6-ORCA2
version, with a horizontal resolution 2º longitude and 0.3º to 1.5º latitude, and 31 vertical levels. It is forced by daily
meteorological data from NCEP reanalysis (Kalnay et al., 1996) over the period 1948-2020, with output for 2009-
2013 used here. This model version is identical to that used to estimate the ocean CO$_2$ sink in the Global Carbon
Budget 2021 annual update (Friedlingstein et al. 2021).

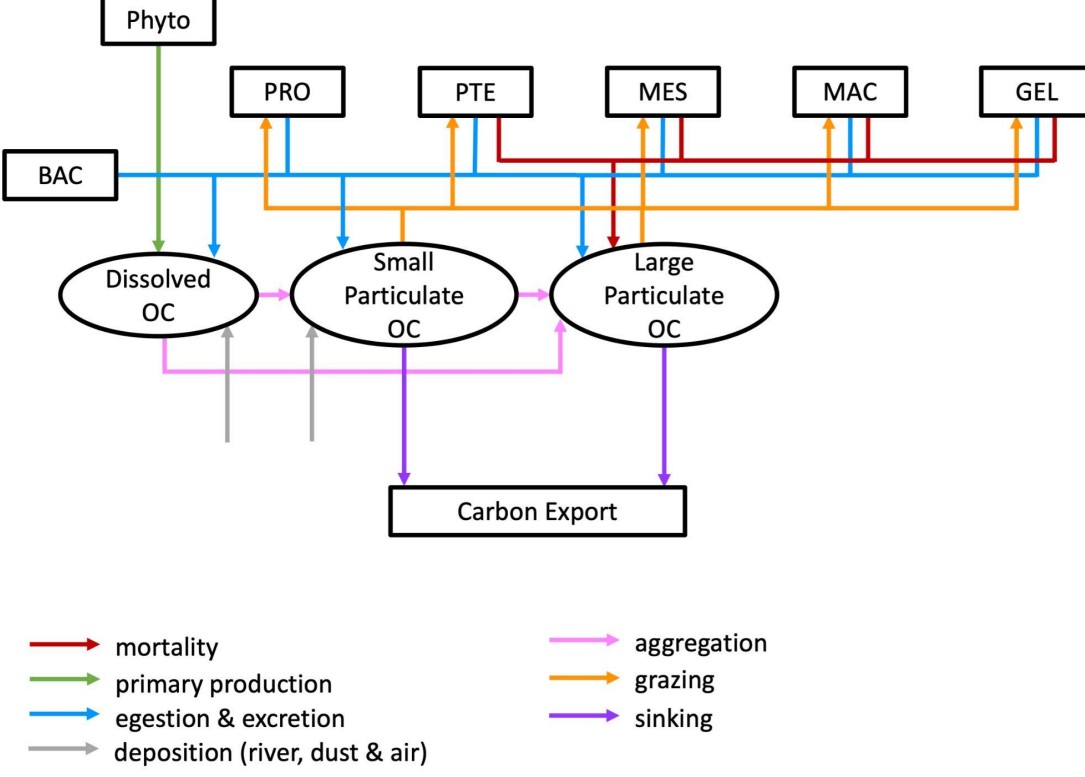


**Figure 2. Schematic representation of the flow of matter in and out of the two particulate organic carbon (OC) components**
**of the PlankTOM12 marine ecosystem model. The various boxes represent: Phyto - phytoplankton that includes diatoms**
**(DIA), mixed phytoplankton (MIX), coccolithophore (COC), picophytoplankton (PIC), phaeocystis (PHA) and N$_2$-fixers**
**(FIX); PRO - protozooplankton, PTE - pteropod, MES - mesozooplankton, MAC - macrozooplankton, GEL - gelatinous**
**zooplankton, BAC - bacteria.**
PlankTOM12 represents ecosystem dynamics based on the representation of 12 PFTs: diatoms (DIA), mixed
phytoplankton (MIX), coccolithophore (COC), picophytoplankton (PIC), phaeocystis (PHA), N$_2$-fixers (FIX), micro-
or protozooplankton (PRO), pteropod (PTE), mesozooplankton (MES), gelatinous zooplankton (GEL), and bacteria
(BAC). PlankTOM12 keeps track of the carbon biomass (µmol/L) of these PFTs over model depth levels resulting
from environmental and ecosystem processes and their interactions (Le Quéré et al. 2016).





PlankTOM12 represents sinking processes through the explicit representation of two organic particle of different size,
with small particles sinking at a constant speed of 3 m/d, and larger particles sinking at a variable speed between 3
and 150 m/d depending on the ballast effect of their mineral content (Buitenhuis et al., 2013). In addition, a dissolved
organic carbon component is transported via ocean currents. Particles are generated through mass flux from the PFTs
resulting from mortality and egestion and from aggregation through differential sinking or turbulent coagulation, and
destroyed through grazing by zooplankton and remineralisation by bacteria and through disaggregation from shear
currents. Large PFTs contribute mostly to $POC_L$, while small PFTs contribute mostly to $POC_S$. (Le Quéré et al. 2016;
Fig. 2).
The NEMO-PlankTOM12 model output was sampled at the time and location identified from the observations
mentioned above to create a synthetic dataset. The model grid-coordinate closest to the real geographical position was
chosen. If several measurements were co-localised at the same grid coordinate and same time step (day for daily
PlankTOM12 and month for monthly PlankTOM12 outputs), it is counted as one measurement. This model sampling
produced 400 positions when using the daily or monthly PlankTOM12 outputs. All drivers and targets were taken
from the model output at the corresponding coordinates up to 1400 m depth. These outputs served as the reference for
validation and evaluation of the ML methods and for establishing the sets of the most important drivers.
**2.2. Method.**
We tested 2 ML methods that are widely used in target's reconstruction based on tabular data sets: the Random Forest
regressor and the XGBoost (Extreme Gradient Boosting) regressor. The Random Forest (RF) regressor is an ensemble
algorithm that contains a number of decision trees on various subsets of the given dataset and takes as output the
average of prediction from each tree estimator. RF can run several trees at the same time allowing a use of a large
number of input variables, and it is robust to overfitting (Biau, 2012). XGBoost (XGB) regressor is an effective tree-
based ensemble learning algorithm (Chen and Guestrin, 2016). It builds several models sequentially where each new
model attempts to correct errors from the previous one. XGBoost uses the gradient descent algorithm to minimise the
loss function of the model. Using RF and XGBoost we can estimate the driver importance to identify which driver has
the greatest impact on the predictions. To check the driver importance, we use drop_col_feat_imp python function
(https://gist.github.com/erykml/6854134220276b1a50862aa486a44192). This method estimates how the accuracy of
the ML output changes if one of the drivers is dropped off from a driver set (DS) based on the training dataset.
Effective ML algorithm requires sets of training, validation and test data. The training data builds up the ML model.
Model evaluates training data repeatedly to learn about the relationship between inputs (driver set) and known outputs
(target set) and adjusts itself to better represent the target. The purpose of validation data is to evaluate the model
during its training by introducing new unseen data. It allows us to evaluate how a developed model works on a new
dataset and to optimise hyperparameters. The test data evaluate the final accuracy of the ML model and confirm that
the model works correctly on any unseen data. It is new data that did not participate in the training algorithm. The
accuracy is worse on validation and test data compared to training data set. The difference in model performance on
training and validation data can signal an overfitting, while this difference between validation and test data can
demonstrate an effect of data mismatch. It is worth noting that RF does not necessarily need validation data set as they
perform internal validation. During the training algorithm each tree is constructed from a random subset of original
data, usually it represents two thirds of data and one third of data is used to estimate out-of bag error to assess model
performance. XGB uses a validation data set to evaluate the model during training and to prevent overfitting by
applying an early stopping. In the present study the available data were split into training and validation data sets (Fig.
3a). Validation data is not included in RF training, however we use it to test the performance of trained RF and tune
hyperparameters afterwards. The test data are taken from the regions where there are no observations (Fig. 3b): 3
months for each year from the period 2009-2013 and 6 positions for each month were chosen randomly. This will
allow us to identify the possible accuracy of reconstruction that can be reached in these regions when we will apply a
developed method to real observations. However, when $POC_S$ and $POC_L$ will be reconstructed using only real-world
observations, we will need to split all available data into training, validation and test data sets.



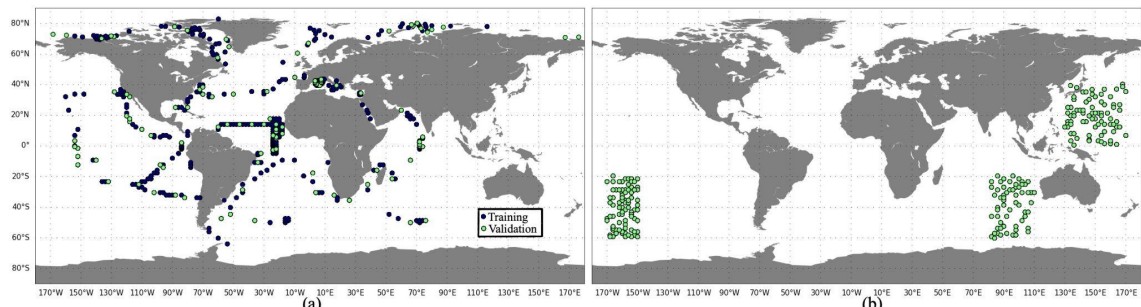

(a)    (b)

**Figure 3. The spatial distribution of: (a) - training (blue) and validation (green) data sets; (b) - test data set; based on**
**PlankTOM12 monthly outputs.**
We use RandomForestRegressor function from scikit-learn (https://scikit-
learn.org/stable/modules/generated/sklearn.ensemble.RandomForestRegressor.html) with its default parameters and
min_sample_leaf equals 20. To apply XGBoost regressor we use XGBRegressor from xgboost
(https://xgboost.readthedocs.io/en/stable/python/python_intro.html). Parameters were set as follows
n_estimators=2000, max_depth=7, eta=0.01, subsample=0.7, colsample_bytree=0.8, gamma=0.01 for $POC_L$ and
gamma= 0.3 for $POC_S$, early_stopping_rounds = 10.
We tested 27 driver sets (DSs) that are summarised in Table 1. For each DS we identify the most important drivers
that influenced the reconstruction of small ($POC_S$) and large ($POC_L$) particulate organic carbon concentration. The
drivers include geographic variables (depth, sin(latitude), cos(longitude)), physical variables (incident light, MLD,
co-located temperature), chemical variables ($PO_4$, $NO_3$, including co-located values and averages over the MLD), and
biological variables (chlorophyll $a$, 12 PFTs listed above: DIA, MIX, COC, PIC, PHA, FIX, PRO, PTE, MES, GEL,
BAC, including co-located values and averages over the MLD).
**Table 1. Compounds of driver's sets: dark grey cells correspond to the drivers present in the driver set. 'vp' – vertical**
**profile, 'mean' – average over MLD, 'back' – values from previous month.**

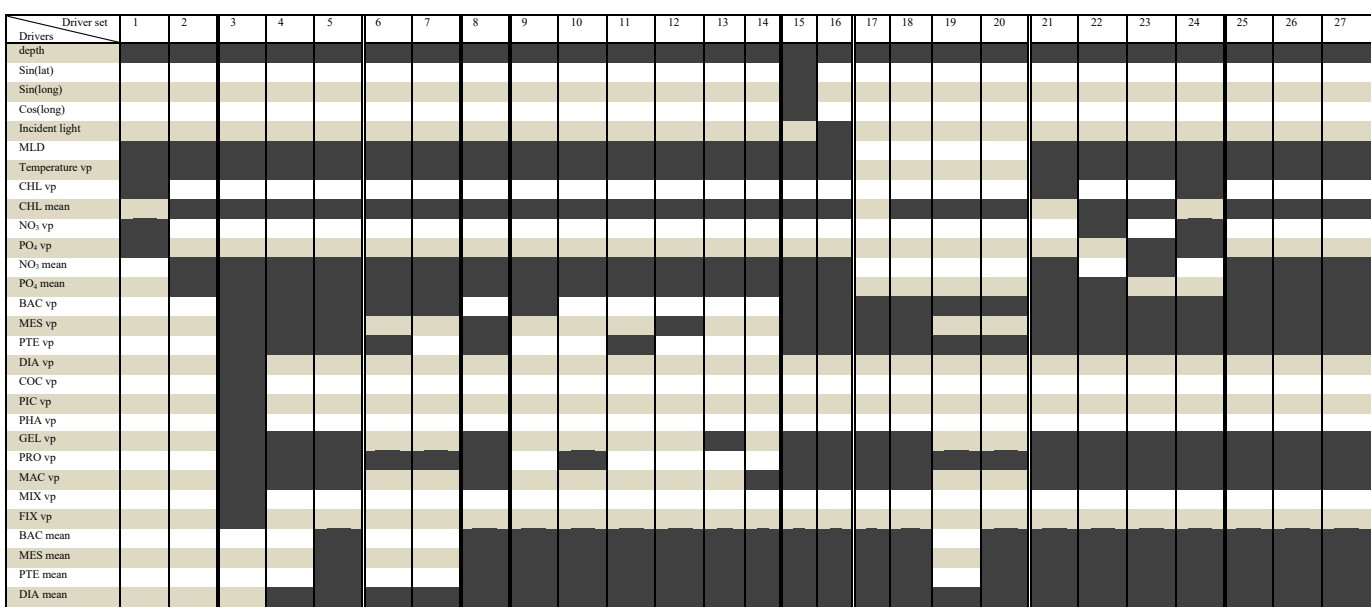






The driver sets can be split into 9 thematic groups which together test the role of PFTs and sub-classes within, the role of surface versus depth profiles for some variables, and the role of information from the previous month:

I. No PFTs (short name (sh.n.) 'No PFT'): Driver sets 1 and 2 do not include any PFTs and focus on the influence of temperature, MLD, chlorophyll *a*, $NO_3$ and $PO_4$ on $POC_S$ and $POC_L$ reconstruction.

II. Introduction of PFTs (sh.n. 'PFT introduction'): DSs 3, 4 and 5 are dedicated to the investigation of the introduction of PFTs in the reconstruction. In DS 3 we introduced 12 PFTs vertical profiles, even though this information will be challenging to reproduce with observations due to the lack of data. Nevertheless, it is important to test the capacity of ML if all 12 PFTs were available over the depth. DS 4 includes the vertical profiles of 6 heterotrophs (zooplanktons and bacteria) because they contribute to influencing the vertical distribution of $POC_S$ and $POC_L$, and 6 phytoplankton averaged over MLD because they are responsible for primary production. In DS 5 we added averages over MLD of the 6 heterotrophs that were not included in DS 4.

III. Big zooplankton (sh.n. 'Zooplankton combined'): In DSs 6 and 7 we tested the influence of big zooplanktons summed into one variable to account for their combined effect rather than the distinctions among PFTs. The big zooplankton is represented by the sum of mesozooplankton, gelatinous zooplankton and macrozooplankton in DS 6, with the addition of pteropod in DS 7.

IV. Exclusion of bacteria (sh.n. 'No vertical BAC'): DS 8 does not have a bacteria (BAC) vertical profile compared to set 5.

V. Individual zooplankton types (sh.n. 'Individual PFT'): DSs 9, 10, 11, 12, 13 and 14 test the influence of individual types of heterotrophs, bacteria (BAC), microzooplankton (PRO), pteropod (PTE), mesozooplankton (MES), gelatinous zooplankton (GEL), microzooplankton (MAC), respectively.

VI. Geographical position and seasons (sh.n. 'Lat-Long' and 'Incident light'): DS 15 is based on DS 5 (which showed the most promising results) and includes geographical coordinates as additional drivers in the form of sin(lat), sin(long), cos(long). DS 16 includes in addition to the DS 5 the role of incident light.

VII. Use of only PFTs and chlorophyll *a* (sh.n. 'PFT only + CHL'): DS 17 is based on only the 12 PFTs, while DS 18 is formed from DS 17 plus information on chlorophyll *a* averaged over the MLD. DSs 19 and 20 are based on DS 6. To form the DS 19 we exclude temperature, $NO_3$ and $PO_4$ from the list of drivers in DS 6. DS 20 is an extended version of DS 19 with all 12 PFTs concentration averaged over the MLD.

VIII. Chlorophyll *a* and chemical variables (sh.n. 'Biochemical variables'): DSs 21, 22, 23, 24 are based on DS 5 and test the individual influence of chlorophyll *a* (DS 21), $NO_3$ (DS 22), $PO_4$ (DS 23) vertical profiles and its ensemble (DS 24).

IX. Previous time step (sh.n. 'Month - 1'): DSs, 25, 26 and 27 investigate the role of chlorophyll *a* (DS 27) and some zooplanktons from the previous time step: gelatinous zooplankton and microzooplankton (DS26); gelatinous zooplankton, micro- and macrozooplankton, averaged over MLD chlorophyll *a* and coccolithophore (DS25).

The evaluation of the method is based on the mean correlation coefficient, total root-mean square errors (RMSE), and total absolute bias between the ML outputs and PlankTOM12 $POC_S$ and $POC_L$ components. Moreover, we provide the global maps of correlation coefficient and RMSE to vertical profiles of $POC_S$ and $POC_L$ at each grid point. Global maps help to identify zones where the large errors can be hidden in the mean diagnostics due to the error compensation.

**3. Results.**



257        **3.1. Data analysis.**

In this study we test the capacity to reconstruct particulate organic carbon from sparse observations by using ML and
a synthetic data set based on the PlankTOM12 model output. We compare observations and the output of the ocean
model to provide a minimum of validation for the model data and to help explain differences in ML results when
applied to real observations in the future.
Figure 4 shows the vertical profile of small ($POC_S$) (Fig.4a) and large ($POC_L$) particulate organic carbon (Fig.4b)
based on the median from observations (green) and from daily PlankTOM12 model output (blue). Shading
corresponds to values between 0.25 and 0.75 percentiles.

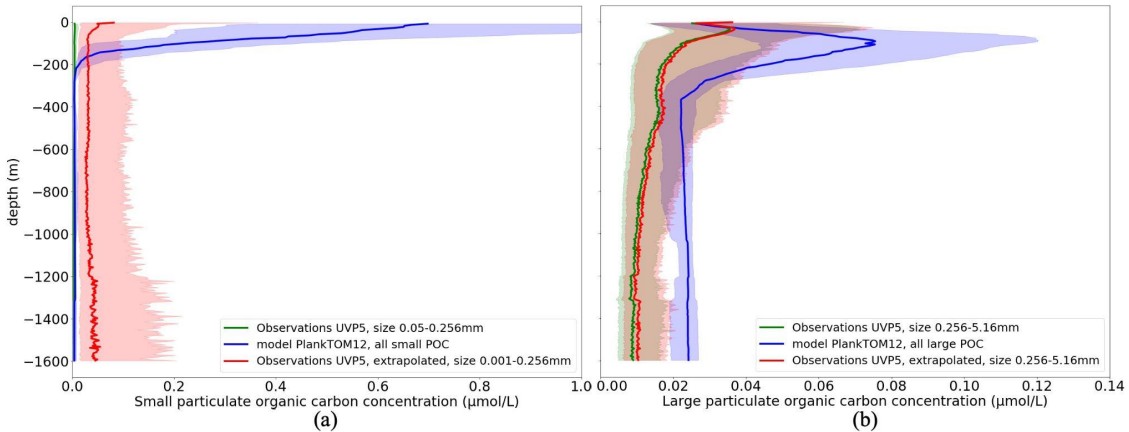

**Figure 4. Comparison of the vertical distribution of particulate organic carbon concentrations (µmol/l) from UVP5**
**measurements (green), PlankTOM12 daily model (blue) and extrapolated UVP5 measurements (red): (a) - small particulate**
**organic carbon concentrations; (b) - large particulate organic carbon concentrations. The median is shown in dark and the**
**shading corresponds to values between the 0.25 and 0.75 percentiles. The size of the particles does not correspond**
**completely between the observations and the model, for $POC_L$ the UVP particle range is chosen as 0.256-5.16 mm that**
**corresponds approximately to the $POC_L$ in the model.**
PlankTOM12 overestimates $POC_S$ up to 3 µmol/L in the first 200m (Fig.4a, green and blue curves). UVP5 does not
capture all small particles that is why we extrapolated the size range of UVP measurements (red curve, see details in
2.1.1). The extrapolated measurements show an increase in $POC_S$ in the first 100m, however this increase still results
in the lower concentration compared with PlankTOM12. These results indicate that PlankTOM12 overestimates the
concentration of small particulate organic carbon. PlankkTOM12 also overestimates $POC_L$ by up to 0.08 µmol/L in
the first 200m and does not catch the increase in $POC_L$ between 300 and 500m. Observations show an increase in
$POC_L$ concentration in the first 50m while PlankTOM12 reproduces it lower, at 100m. The RMSE between modelled
and observed $POC_S$ is 0.33 µmol/L, with correlation coefficient equals 0.083. RMSE equals 0.23 µmol/L with
correlation coefficient 0.061 for $POC_L$. The exclusion of isolated large values of $POC_L$ (>2 µmol/L) from the
observation data set reduces the RMSE of $POC_L$ to 0.062 µmol/L with correlation 0.18. We believe that these
differences result from differences in space and time resolution of observations and ocean model outputs. *In-situ*
measurements are obtained at a particular time of the day and a particular latitude-longitude position while the model
provides estimations over the day (or month) and on the model grid (2º longitude and mean 1.1º latitude resolution).
We concluded that observed and modelled $POC_S$ and $POC_L$ have a common tendency in their vertical distributions.
However, among other things, differences in amplitudes may affect our findings in this work when we develop a ML
method based on observations only.
Due to the constraint in data availability further we use monthly PlankTOM12.
Before developing a ML method, we investigate the interactions between targets and drivers in the model. Table 2
shows the correlation coefficients between the $POC_S$ and $POC_L$ and corresponding drivers that can influence $POC_S$
and $POC_L$ variability. $POC_S$ correlates with gelatinous zooplankton (GEL, r=0.66), microzooplankton (PRO, r=0.63),



coccolithophore (COC, r=0.56), as well as with their values from previous time step (GEL, r=0.67; PRO, r=0.51;
COC, r=0.59). Coccolithophore is one of the most abundant phytoplankton types in this version of the PlankTOM
model (similar to Wright et al., 2021). The growth of phytoplankton transfers dissolved inorganic carbon into dissolved
organic carbon which further aggregates into $POC_S$ and $POC_L$. Also, $POC_S$ is generated from microzooplankton
egestion and excretion (Fig. 2). In addition to the mentioned above PFTs, $POC_S$ shows a correlation 0.44 with
temperature vertical profile at both the considered time step and at the previous time step. $POC_S$ has a negative
correlation with $NO_3$ (r=-0.46) and $PO_4$ (r=-0.41).
$POC_L$ does not show a high correlation with any of the proposed drivers individually and is therefore most likely the
result of multiple processes and/or multiple drivers, including for its production and destruction. The ML approach
should be able to identify combinations of drivers beyond straight correlations that are investigated directly here.
$POC_L$ has the highest correlation with chlorophyll *a* (r=0.42), gelatinous zooplankton at the considered time step
(r=0.37), and at previous time step (r=0.36). Gelatinous zooplankton contribute to $POC_L$ formation through egestion
and excretion mainly from mucus (Fig. 2). As explained in Wright et al. (2021), mucus forms a large low-density mass
through aggregation with other particles. It can explain a correlation of gelatinous zooplankton with $POC_L$ in
PlankTOM12.
**Table 2. Correlation coefficient between small ($POC_S$) and large ($POC_L$) particulate organic carbon concentration and**
**possible drivers. Estimation is based on monthly PlankTOM12 output at the position of real-world observations from Fig.**
**1. 'vp' – vertical profile, 'mean' – average over MLD, 'back' – values from previous month.**

| Driver | $POC_S$ | $POC_L$ | Driver | $POC_S$ | $POC_L$ | Driver | $POC_S$ | $POC_L$ |
|---|---|---|---|---|---|---|---|---|
| **POC** | 1.00 | 0.33 | BAC vp | -0.14 | 0.15 | BAC back vp | -0.10 | 0.09 |
| **GOC** | 0.33 | 1.00 | MES vp | -0.09 | 0.07 | MES back vp | -0.09 | -0.07 |
| **Depth** | -0.32 | -0.24 | PTE vp | -0.07 | 0.17 | PTE back vp | -0.08 | 0.08 |
| **Temperature vp** | **0.44** | 0.17 | DIA vp | -0.04 | 0.15 | DIA back vp | -0.03 | 0.09 |
| **Temp back vp** | **0.44** | 0.17 | COC vp | **0.56** | 0.31 | COC back vp | **0.60** | 0.31 |
| **MLD** | -0.01 | -0.07 | PIC vp | 0.00 | 0.07 | PIC back vp | 0.06 | 0.06 |
| **$NO_3$ vp** | **-0.46** | 0.01 | PHA vp | 0.27 | 0.15 | PHA back vp | 0.30 | 0.17 |
| **$PO_4$ vp** | **-0.41** | 0.04 | GEL vp | **0.66** | **0.37** | GEL back vp | **0.68** | **0.36** |
| **$NO_3$ back vp** | **-0.46** | 0.03 | PRO vp | **0.63** | 0.16 | PRO back vp | **0.51** | 0.14 |
| **$PO_4$ back vp** | **-0.41** | 0.05 | MAC vp | 0.07 | 0.14 | MAC back vp | 0.08 | 0.13 |
| **CHL vp** | 0.18 | **0.42** | MIX vp | 0.07 | 0.17 | MIX back vp | 0.03 | 0.05 |
| **CHL back vp** | 0.11 | 0.22 | FIX vp | -0.00 | 0.23 | FIX back vp | -0.00 | 0.23 |


**3.2. Development of the Machine Learning method.**



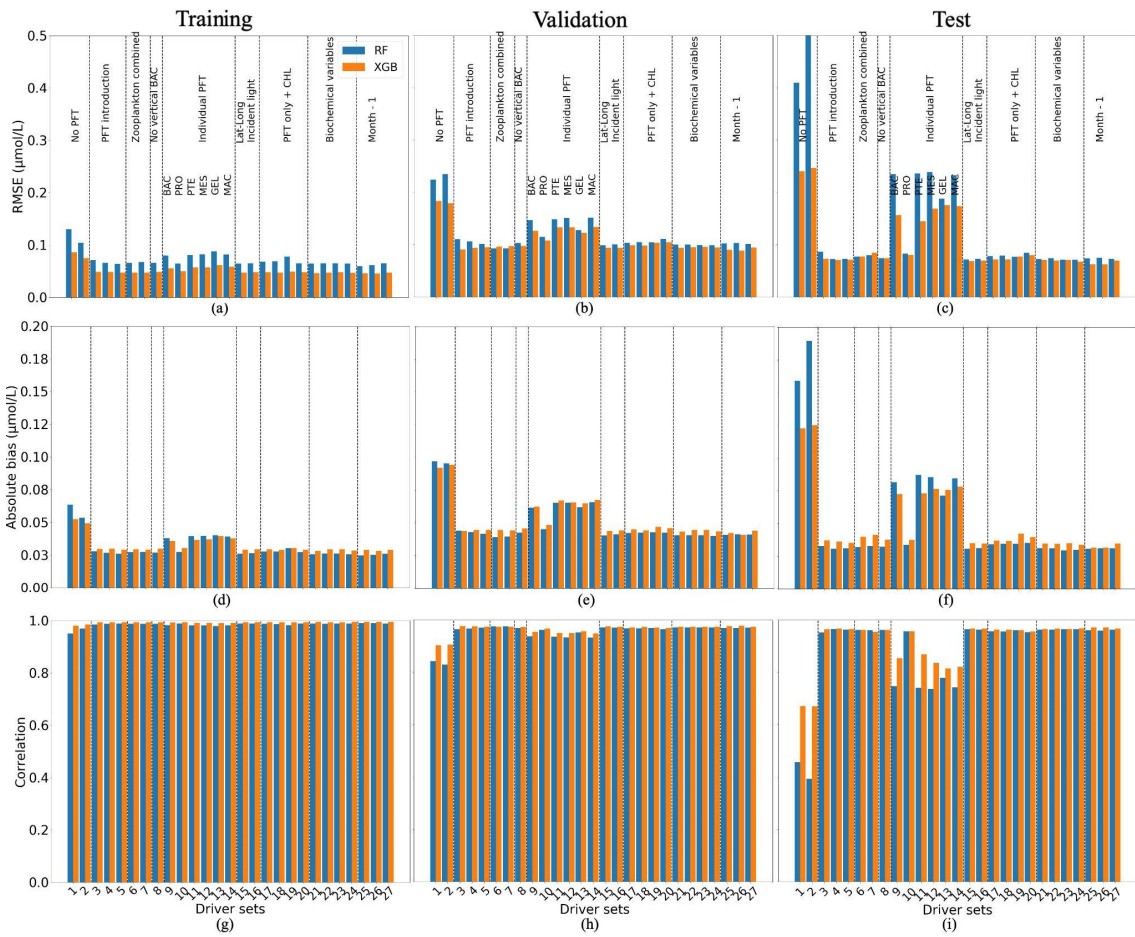

**Figure 5. Comparison of the performance of the Random Forest (RF) and XGBoost methods and their fit to data for small (POC$_S$) particulate organic carbon concentration; (a, b, c) - RMSE in μmol/l, (d, e, f) - absolute bias in μmol/, (g, h, i) - correlation coefficient; (a, d, g) - training data set, (b, e, h) - the validation data set, (e, f, i) - the test data set. Results compare data from the original (sampled) PlankTOM12 model output and POC$_S$ reconstructed using RF (blue) and XGB (orange). The low RMSE and absolute biases indicate better performance of the ML method.**

We tested 27 sets of drivers (Table 1) and two ML methods, Random Forest (RF) and XGBoost regression (XGB).

Figure 5 shows the statistics of POC$_S$ reconstruction using RF and XGB. XGB (orange) generally overperforms RF (blue). The statistics are slightly worse for the validation and test data sets, as expected. For reconstructions using XGB, the RMSE and absolute bias are about 0.05 μmol/L and 0.03 μmol/L on the training data set and vary around 0.1 μmol/L and 0.05 μmol/L, on the validation and test data, respectively. Correlation coefficients (Fig. 5g, h, i) have high values on all datasets showing that the vertical profiles of POC$_S$ have a correct shape. These results show that the available spatial and temporal coverage of *in situ* observations can be sufficient to reconstruct POC$_S$ with an appropriate accuracy over the global ocean. The analysis of global maps (shown below) will help to identify areas with low accuracy and their differences with training regions.

The worse results (highest RMSE, highest absolute bias, lowest correlation) are produced when there are no PFTs in the driver set (DS1 and DS2; Figure 5): for XBoost, RMSEs are 0.24 μmol/L, absolute biases equal to 0.12 μmol/L with correlation coefficient 0.67 on the test data sets. Poor results are also obtained for DS9, 11, 12, 13 and 14: these 5 driver sets do not have any information on microzooplankton (PRO) and show high RMSEs and absolute biases,





around 0.16 μmol/L and 0.074 μmol/L, with low correlation, 0.83, compared with other driver sets which include PRO. These results indicate that microzooplankton plays an important role in POC$_S$ variability in the PlankTOM12 model.

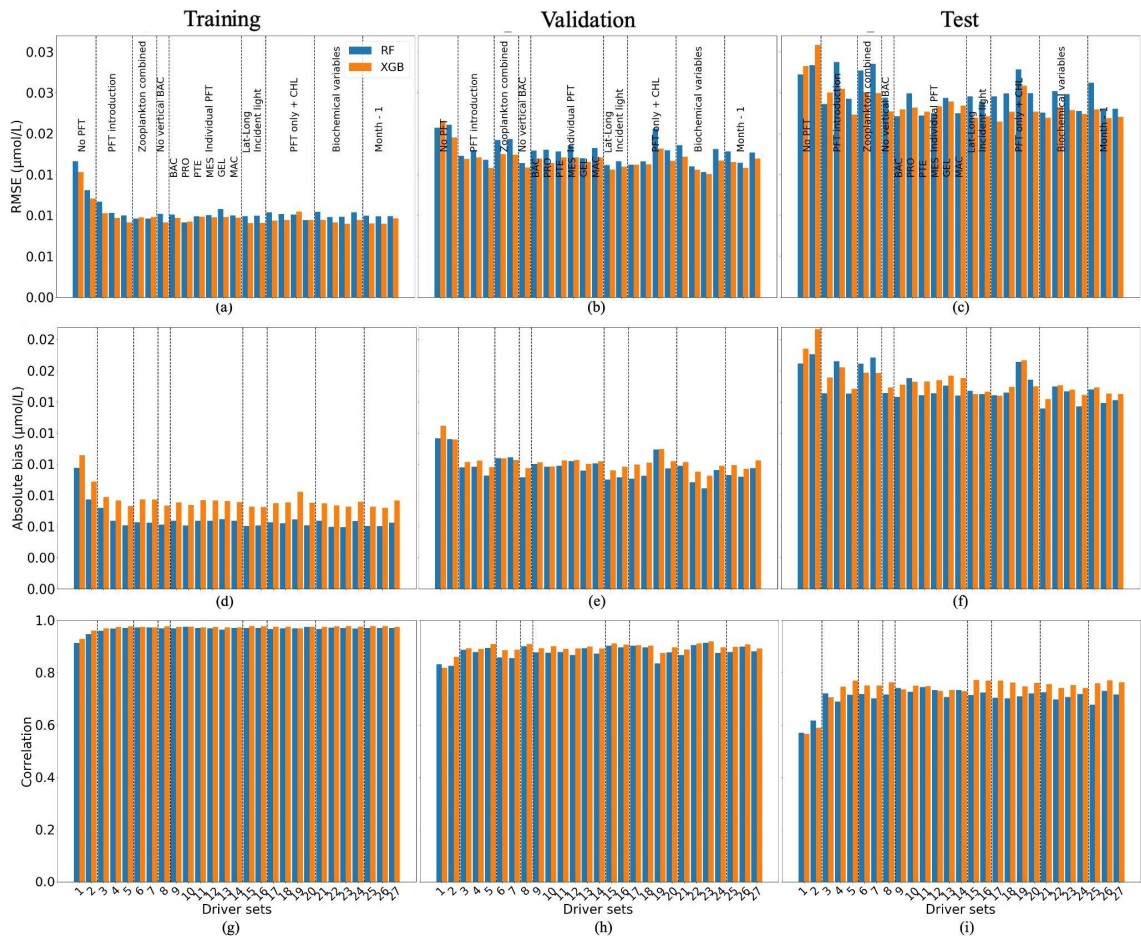

**Figure 6. Comparison of the performance of the Random Forest (RF) and XGBoost methods and their fit to data for large (POC$_L$) particulate organic carbon concentration; (a, b, c) - RMSE in μmol/l, (d, e, f) - absolute bias in μmol/l, (g, h, i) - correlation coefficient; (a, d, g) - training data set, (b, e, h) - the validation data set, (e, f, i) - the test data set. Results compare data from the original (sampled) PlankTOM12 model output and POC$_L$ reconstructed using RF (blue) and XGB (orange). The low RMSE and absolute biases indicate better performance of the ML method.**

Figure 6 shows the statistics of POC$_L$ reconstruction using RF and XGB. XGBoost again slightly overperforms RF on most driver sets. Results for driver sets with PFTs show lower RMSEs and absolute biases, and higher correlation coefficients. Except for the effect of PFTs on the POC$_L$ reconstruction, we did not observe a clear influence of one driver or group of drivers. Using XGBoost the reconstruction of POC$_L$ shows the RMSE in DS1 is high at 0.03 μmol/L, while it is in the range of 0.021-0.026 μmol/L in DS3-DS27, with absolute bias in DS1 of 0.02 μmol/L and 0.015-0.018 μmol/L for DS3-DS27 based on test data (Fig. 6c, f). Likewise, a correlation coefficient of 0.56 for DS1, and between 0.7 and 0.77 for DS3-DS27 based on the training data set (Fig. 6g).

We estimated the ranking of importance for each driver averaged over 27 driver sets (Table 1) for RF and XGB (Fig. 7). Both, RF and XGB, show that microzooplankton (PRO), depth level, temperature, NO$_3$ and PO$_4$ play a dominant role in reconstruction of POC$_S$. The absence of gelatinous zooplankton (GEL) can slightly improve the reconstruction.





Also, latitude and longitude do not affect POC$_S$ reconstruction. The depth level, temperature, MLD, microzooplankton
(PRO) and phaeocystis (PHA), PO$_4$, and chlorophyll *a* averaged over MLD play a dominant role in POC$_L$
reconstruction.
The sinus of latitude is in the top ten drivers that most affect POC$_L$ using XGBoost method: POC$_L$ distribution depends
on latitude zones. As for POC$_S$, gelatinous zooplankton (GEL) shows a negative rank of driver importance and its
removal from the list of drivers can improve the statistics of reconstruction. Also, chlorophyll *a* concentration from
the previous month shows a similar effect on POC$_L$ (Fig. 7c, d).
Based on Figures 5, 6 and 7 we have chosen 10 driver sets with low RMSEs and absolute biases, and high correlation
coefficients (based on test data set) for POC$_S$ and POC$_L$ to provide global maps of these statistics and to see their
regional distributions. DS 5, 15, 16, 21, 22, 23, 24, 25, 26, 27 were chosen for further investigation of POC$_S$
reconstruction; DS 5, 8, 15, 16, 17, 21, 23, 25, 26, 27 – for POC$_L$ reconstruction. Common for POC$_S$ and POC$_L$ driver
sets 5, 15, 16, 21, 23, 25, 26, 27 include all PFTs and their average over MLD, geographical positions and incident
light as well as chlorophyll *a*, PO$_4$, and gelatinous zooplankton and microzooplankton from the previous time step
(Table 1). Also, we found that POC$_S$ reconstructions rest on biochemical conditions (DSs 21 and 24), while POC$_L$
reconstruction mostly depends on the composition of the PFTs in the driver set (DSs 8 and 17). Additionally, we keep
DS1 to demonstrate a global effect of PFTs on reconstruction.

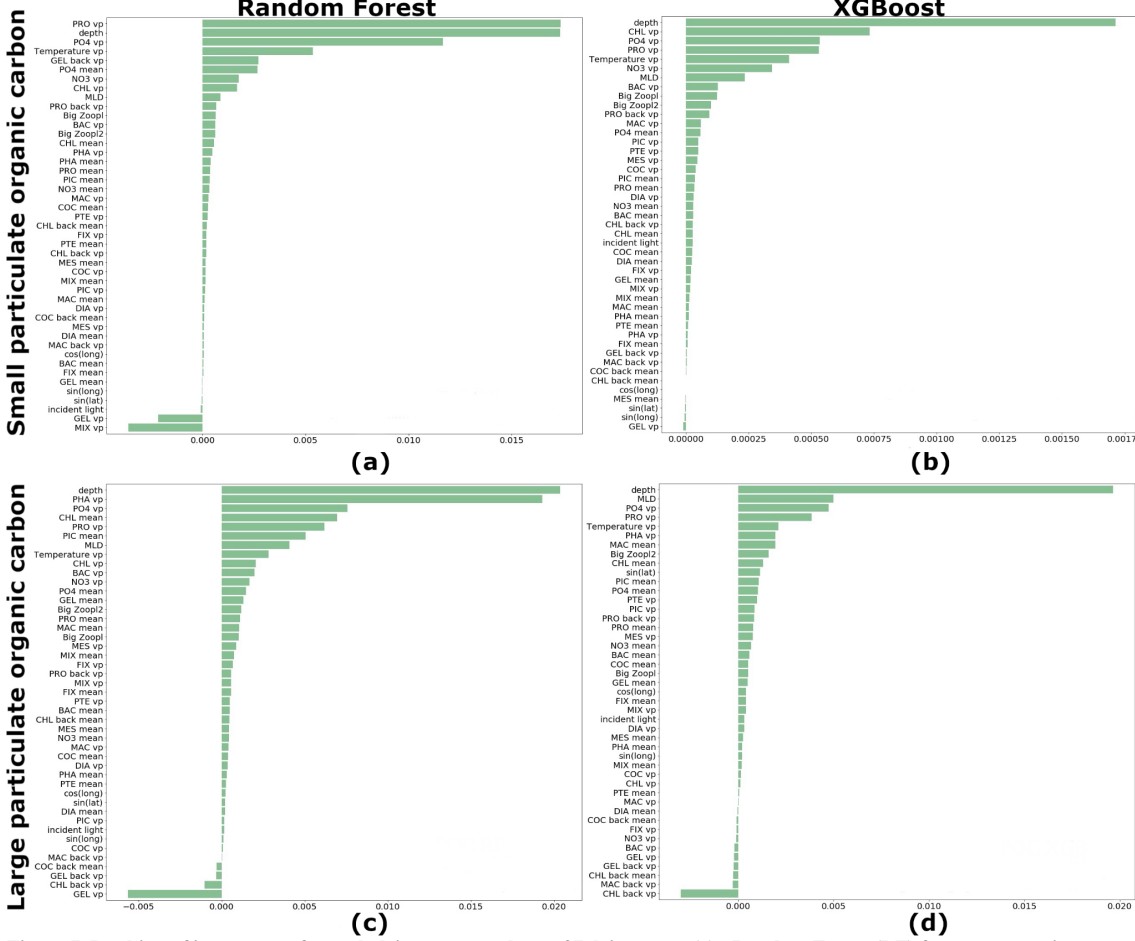

**Figure 7. Ranking of importance for each driver averaged over 27 driver sets: (a) - Random Forest (RF) for reconstruction**
**of small (POC$_S$) particulate organic carbon concentration; (b) - XGBoost (XGB) for small (POC$_S$) particulate organic**





carbon concentration; (c) - RF for POC$_L$ concentration; (d) - XGB for POC$_L$ concentration. 'vp' – vertical profile, 'mean'
– average over MLD, 'back' – values from previous month.

372        **3.3.  POC$_S$ and POC$_L$ vertical profile reconstruction over the global ocean**

In the previous section we showed that XGBoost provides the best results for the reconstructions of POC$_S$ and POC$_L$.
Further we use this ML method. Here we will discuss the regional results of DS1 without PFTs and 10 best driver sets
chosen for each target separately.

**Figure 8. Total averaged over the depth and period 2009-2013 small (POC$_S$) and large (POC$_L$) particulate organic carbon
concentration: (a) – PlankTOM12 POC$_S$, (b) – PlankTOM12 POC$_L$, (c) – reconstruction of POC$_S$ based on DS1 (NoPFT)
using XGBoost, (d) – reconstruction of POC$_L$ based on DS1 using XGBoos, (e) - reconstruction of POC$_S$ based on DS25
(vertical profiles of zooplanktons, and zooplankton and phytoplankton averaged over MLD) using XGBoost, (f) -
reconstruction of POC$_L$ based on DS25 using XGBoost.**

Figure 8 shows POC$_S$ and POC$_L$ concentration averaged over the depth and period 2009-2013 for PlankTOM12 (Fig.
8a, b), XGboost reconstruction based on DS1 (Fig. 8c, d) and XGBoost reconstruction based on DS25 (Fig.8 e, f).





XGBoost captures well the spatial patterns: the high concentration of $POC_S$ in the Equatorial Eastern Pacific and its
low concentration at high latitudes, as well as the high concentration of $POC_L$ in the Equatorial Eastern Pacific and in
the North of the Indian Ocean and its low concentration in the Subtropical North and South Atlantic and in the
Subtropical North Pacific. The presence of PFTs in driver sets (Fig. 8e, f) improves the reconstruction: the spatial
patterns and its amplitude are visually close to ones from PlankTOM12 (Fig. 8a, b). The high concentration of $POC_S$
in the Equatorial Eastern Pacific is represented better using DS25 compared with DS1 where the concentration in the
latitude band 0ºS-20ºS along the Peru is overestimated. Also, small decreases of $POC_S$ in the Subtropical North and
South Atlantic are captured better when we use DS25. Similar for $POC_S$, the high concentration in the Equatorial
Eastern Pacific is represented better using DS25 compared with DS1 where the concentration misses the small
decrease between 20ºN and 0ºN. Also, small decreases of $POC_L$ in the Subtropical North and South Atlantic as well
as in the Subtropical North Pacific are pronounced better with DS25.
Figure 9 shows regional correlation coefficients and RMSEs between PlankTOM12 and XGBoost reconstruction over
the global ocean for 2009-2013. We averaged correlation coefficient and RMSEs over 7 latitude zones: 90ºN-60ºN,
60ºN-40ºN, 40ºN-20ºN, 20ºN-20ºS, 20ºS-40ºS, 40ºS-60ºS, 60ºS-90ºS. In $POC_S$ reconstruction, the DS1 shows the
lowest correlation across latitude bands (between 0.22 and 0.9), and highest RMSEs (0.05-0.34 µmol/L; Fig.9a, b).
DSs 25 and 26 show the highest correlations in the range of 0.68 (in region 60ºS-90ºS) and 0.97 (in region 20ºN-20ºS)
and the lowest RMSEs in the range of 0.021 (in region 60ºS-90ºS) and 0.14 µmol/L (in region 90ºN-60ºN). DS25
contains information on the previous-month distribution for micro-, macrozooplankton and gelatinous zooplankton
vertical profiles as well as coccolithophores and chlorophyll *a* averaged over the MLD. DS26 is like DS25 but the
drivers which bring information from the previous month are microzooplankton and gelatinous zooplankton vertical
profiles.
10 driver sets (excluding DS1) show their highest RMSEs in $POC_S$ reconstruction in the region 90ºN-60ºN, with
values up to 0.14 µmol/L in DS27 (Fig. 9b). Figure 10 shows maps of RMSEs (a, b) and correlation coefficients (c,
d) between PlankTOM12 and reconstructed small particulate organic carbon ($POC_S$) by XGBoost using driver sets 1
(a, c) and 25 (b, d). The region 90ºN-60ºN shows improvement in RMSEs and absolute biases in DS25 compared with
DS1, with RMSEs decreasing from 0.2 µmol/L to 0.03 µmol/L in Norwegian Sea, Baffin Bay, and the Arctic Ocean.
However, errors stay high in the coastal regions, Northwestern passage and Hudson Bay that contribute to the high
total RMSEs in this region. Results are similar for the region 60ºN-40ºN, where correlation coefficients increased
from 0.3 to 0.87 on average over these zones (Fig. 10c, d). The tropical region 20ºN-20ºS shows correlation coefficient
up to 0.97 for all driver sets except DS1. However, RMSEs are high in the tropical region, about 0.11µmol/L on
average (Fig. 9b), with RMSEs values of 0.2 µmol/L in the Tropical Eastern Pacific and Bay of Bengal in DS25 (Fig.
10b). The high RMSEs in the Tropical Eastern Pacific can indicate insufficient data in a region of high interannual
variability to correctly reconstruct $POC_S$ distribution. The region of the Southern Ocean (>60ºS) shows the lowest
correlation coefficients (in the range of 0.64-0.69) and RMSEs (in the range 0.023-0.044 µmol/L) for $POC_S$ (Fig. 9a,
b). The inclusion of PFTs in the driver set significantly improves the RMSE in the region around 40ºS for small
($POC_S$) particulate organic carbon. The statistics are improved by about 75% in the region 40ºS-60ºS with RMSE
decreasing from 0.18 (DS1) to 0.03 (DS25) and the correlation coefficient increasing from 0.22 (DS1) to 0.84 (DS25),
on average (Fig. 9a, b; Fig. 10). The improvements in the Southern region are related to the role of zooplankton in the
carbon flux in this area (Le Quéré et al., 2016; Wright et al., 2021).







**Figure 9. Correlations and RMSE averaged over latitude zones between PlankTOM12 and XGBoost reconstruction over**
**the global ocean for 2009-2013: (a, c) - correlation coefficient, (b, d) - RMSE in μmol/l (b, d);. (a, b) - small particulate**
**organic carbon (POC$_S$), (c, d) - large particulate organic carbon (POC$_L$).**
In POC$_L$ reconstruction, DS1 also shows the lowest correlation coefficients (0.35-0.75) and the highest RMSEs (0.027-
0.47 μmol/L) (Fig. 9c, d). DS25 shows the best results on average, with the correlation coefficient varying between
0.43 (in the region 60ºS-90ºS) and 0.84 (in the region 20ºN-20ºS), and RMSE varying between 0.021 (in the region
20ºS-40ºS) and 0.046 (in the region 90ºN-60ºN) μmol/L. POC$_L$ are reconstructed better in subtropical and tropical
regions compared to high latitude zones (Fig. 9c, d).
As for POC$_S$, 10 driver sets (excluding DS1) show their highest RMSEs in POC$_L$ reconstruction in the region 90ºN-
60ºN, with values up to 0.05 μmol/L in DS27 (Fig. 9d). Figure 11 shows maps of RMSEs (a, b) and correlation
coefficients (c, d) between PlankTOM12 and reconstructed large particulate organic carbon (POC$_L$) by XGBoost using



driver sets 1 (a, c) and 25 (b, d). Contrast to POC$_S$ reconstruction, the region 90ºN-60ºN does not show improvement in RMSEs for POC$_L$ reconstruction (Fig. 11b) in DS25 compared with DS1, with still high RMSEs in Norwegian Sea, Baffin Bay, and the Arctic Ocean, and additionally for POC$_L$ in Greenland Sea, where the algorithm did not have data for training. Similar to POC$_S$, errors stay high in the coastal regions, Northwestern passage and Hudson Bay that contribute to the high total RMSEs in this region.

Global maps of statistics suggest that the most sensible region to driver set's composition for POC$_L$ is the Southern Ocean, as for POC$_S$ (Fig. 11). In the 40ºS-60ºS region, RMSE is reduced from 0.037 μmol/L in DS1 to 0.024 μmol/L in DS25 (Fig. 9d), and the correlation coefficient is increased from 0.42 to 0.66 (Fig. 9c) on average, respectively. In the Southern region 60ºS-90ºS, RMSE is reduced from 0.047 μmol/L in DS1 to 0.033 μmol/L in DS25, and the correlation coefficient is increased from 0.33 to 0.42 (Fig. 9c) on average, respectively. The average correlation coefficients in this zone were found to be less than 0.5 in all tests with the highest value 0.5 in DS21. DS21 contains all PFTs and chlorophyll *a* vertical profile as drivers. The RMSE for DS21 in this region is close to the one of DS25, 0.34 μmol/L and 0.33 μmol/L, respectively. It identifies the importance of chlorophyll *a* in the Southern Ocean as driver of POC$_L$ variability.

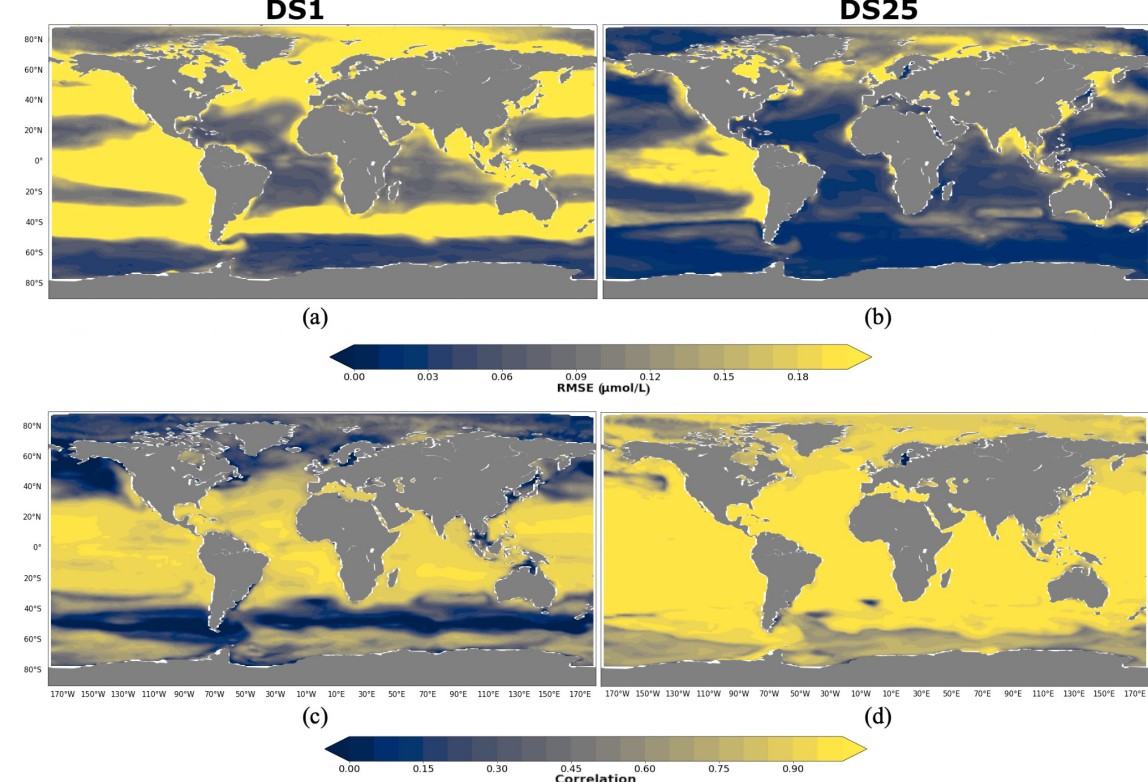

**Figure 10. RMSE and correlation between monthly PlankTOM12 and results of POC$_S$ reconstruction using XGBoost over the period 2009-2013 for POC$_S$. (a, b) – RMSEs, (c, d) – correlation coefficients; (a, c) – reconstruction based on DS1 (NoPFT); (b, d) – reconstruction based on DS25 (vertical profiles of zooplanktons, and zooplankton and phytoplankton averaged over MLD).**



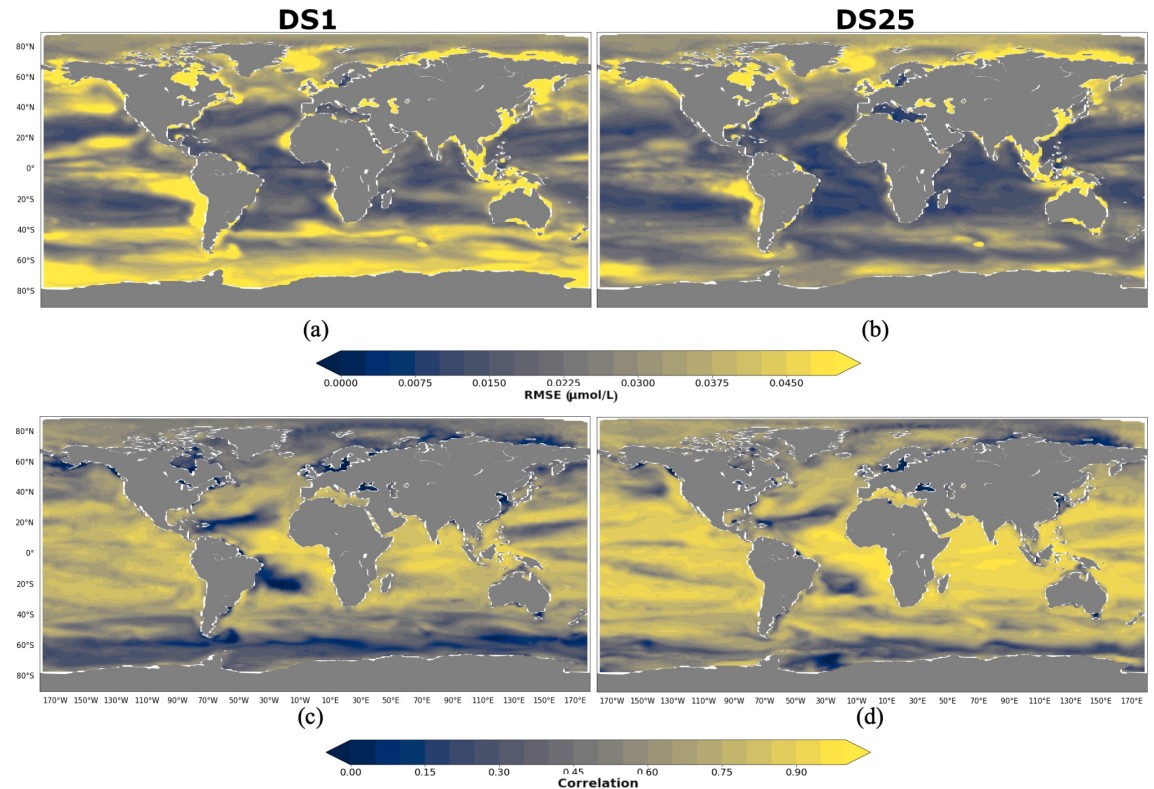

454

**Figure 11. RMSE and correlation between monthly PlankTOM12 and results of POC$_L$ reconstruction using XGBoost over the period 2009-2013 for POC$_L$. (a, b) – RMSEs, (c, d) – correlation coefficients; (a, c) – reconstruction based on DS1 (NoPFT); (b, d) – reconstruction based on DS25 (vertical profiles of zooplanktons, and zooplankton and phytoplankton averaged over MLD).**

The statistics of POC$_S$ and POC$_L$ reconstruction do not vary significantly between driver sets in all regions except in the Southern Ocean. This region is most sensitive to the composition of driver sets for both POC$_S$ and POC$_L$.

**4. Conclusion.**

The aim of this work was to test the potential of using Machine Learning to reproduce modelled concentrations of particulate organic carbon within the ocean using the distribution of available observations. We co-localised outputs of the PlankTOM12 global biogeochemical ocean model with the positions of observations of small (POC$_S$) and large (POC$_L$) particulate organic carbon concentrations. Using PlankTOM outputs as references we could identify the best ML method for POC reconstruction and estimate method's accuracy in regions with poor observational cover.

We tested two ML methods to reconstruct POC$_S$ and POC$_L$: the XGBoost regressor and Random Forest. Both methods are algorithms based on decision trees. XGBoost overperformed Random Forest by about 9% on average for POC$_S$ reconstruction and by about 3% on average for POC$_L$ reconstruction. XGBoost regressor builds the model sequentially improving it at each iterative step. At each iteration, XGBoost regressor analyses the prediction and gives more weight to the data where the fit is still wrong. It is a good tool for an unbalanced data set, like in our case where the data of particulate organic carbon concentration are sparse in time and space.

We tested the influence of a wide range of environmental and ecosystem drivers on POC$_S$ and POC$_L$ reconstruction. The introduction of Plankton Functional Types (PFTs) in the driver set greatly improves the fit and shows a linkage between surface ecosystem structure and particulate organic carbon distribution within the ocean interior. We





improved the accuracy of POC$_S$ reconstruction by 59% on RMSE, 63% on absolute bias and by 52% on correlation
by introducing Plankton Functional Types (PFTs) in the driver sets (from the comparison of DS1 and DS25). The
presence of PFTs in the driver sets also improved the accuracy of POC$_L$ reconstruction by 22% on RMSE, absolute
bias and correlation (from the comparison of DS1 and DS25). POC$_S$ variability mostly depends on the depth level,
vertical profiles of microzooplankton, temperature and PO$_4$. POC$_L$ variability depends on the depth level, MLD,
chlorophyll $a$ averaged over MLD, vertical profiles of temperature, microzooplankton, phaeocystis and PO$_4$.
Additionally, we identified that chlorophyll $a$ in driver sets improves the POC$_L$ reconstruction in the Southern Ocean.
Despite the good accuracy over the global ocean on average, the statistics are worse in the coastal regions and in the
Tropical Eastern Pacific. The coastal regions suffer from the lack of data to represent the coastal dynamics. Therefore
the ML reconstructions assign open-ocean processes to coastal regions, leading to significant biases. The Tropical
Eastern Pacific is a region of strong interannual variability and the sparse measurements in time make it harder to
capture this variability correctly. Other regions with poor coverage by observations - the Eastern Indian Ocean, the
Western Pacific Ocean and the Southern Ocean - show the statistics of reconstruction comparable to one from regions
with a good cover - regions in the Atlantic Ocean. However, we found that the Southern Ocean is a more sensible
region to the driver set's composition. The observational data is particularly sparse in this region and our analysis
suggests that identifying the drivers of importance based on real dataset will be difficult.
Here we showed that the XGBoost regressor and Random Forest are suitable for this problem and can reconstruct
modelled POC$_S$ and POC$_L$ with appropriate accuracy. This is evidenced from the globally averaged correlation
coefficient up to 0.88 for POC$_S$ and 0.68 for POC$_L$, and the globally averaged RMSE up to 20 % (0.08 μmol/L) of
standard deviation of PlankTOM12 POC$_S$, and 65% (0.028 μmol/L) of standard deviation of PlankTOM12 POC$_L$. ML
outputs represent well the spatial patterns of POC$_S$ and POC$_L$ distribution. However, the validity of the approach on
observations is dependent on the availability of co-located information on the drivers of importance. For some drivers
this should be possible (e.g. environmental conditions and chlorophyll $a$), while for other drivers information is more
sparse (e.g. the PFTs). Our analysis suggests that additional PFT observations would help provide broader insights
into the distribution of POC in the ocean. The next step of this work is to apply ML to real data using methods from
the present study.
This study provides insights on the drivers that may be responsible for POC$_S$ and POC$_L$ variability and regional
dependencies. However, the dependencies are simply returning the outcome of complex ecosystem processes among
the drivers as represented in the PlankTOM12 model. Although these processes are based on current understanding
and a broad range of observations (Le Quéré et al., 2016; Wright et al., 2021; Buitenhuis et al., 2019), they remain
results from a model output. Observations could reveal different drivers that are important for POC$_S$ and POC$_L$.
Depending on data availability and its time and space resolution, the final product based on observations should
provide new insights on the drivers that govern particulate organic carbon concentration in the real ocean.
**Data and code availability.** PlankTOM12 data used within this study are available at
https://doi.org/10.5281/zenodo.7324781. UVP5 data can be found at https://doi.org/10.1594/PANGAEA.924375 (R.
Kiko et al., 2021). Codes for data preparation, development of machine learning methods and tests of different driver
sets as well as codes that provide figures shown in the article can be found at https://doi.org/10.5281/zenodo.7326992.
**Author contribution.** All authors contributed to the development of the methodology. ADS, CLQ, ETB designed the
experiments, and ADS carried them out. ADS developed codes and performed the simulations. ADS prepared the
paper with contributions from all coauthors.
**Acknowledgements.** The authors would like to thank Jean-Olivier Irisson for his contribution in the development of
the methodology. ADS, ETB and CLQ acknowledge support from Royal Society (grant RP\R1\191063), and NERC
Marine Frontiers project (grant NE/V011103/1) for CLQ. RK acknowledges support via a "Make Our Planet Great
Again" grant from the French National Research Agency within the "Programme d'Investissements d'Avenir" (grant
no. ANR-19-MPGA-0012) and by the Heisenberg program of the German Science Foundation under project number
522  469175784.

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
