# Peer review of "Testing the reconstruction of modelled particulate organic carbon from surface ecosystem components using PlankTOM12 and Machine Learning"

_Geoscientific Model Development, 2022_

## Author Comment (AC1)

GENERAL ANSWER:

Our manuscript aims to investigate the linkages between surface ecosystems and sinking carbon in almost ideal observational conditions: the case when environmental and ecosystem variables are available at every station where sinking carbon was measured. Although based on model data only, our results suggest some drivers are particularly important (temperature, microzooplankton) and some are less important (vertical profile of gelatinous zooplankton), which will be of value to observationalists. The results will need to be tested with observations before firmly confirming the validity of the drivers. After that the results can help guide observational programs emphasising the variables that must be measured. We also note that a lot of work is ongoing to reconstruct a range of PFT concentrations which will greatly facilitate the application of ML methods such as the one developed here. Please find our detailed answers below. In our reply to the comments raised by both reviewers, we have clarified the purpose of our analysis and expanded the scope of the discussion regarding the applicability of our method to observations.

Answer to Referee #1:

**Referee (R):** Thank you to the authors for this well-written submission that I believe can become a nice step forward.

The authors nicely set up a real world experiment by sampling a model at real world observation locations. However, the authors then utilize drivers that are not always available with at these observation locations (and a very large set of drivers). Thus, the findings about whether ML can be utilized to better understand (and model) the transfer of POC to depth are not as applicable without a substantial change in observations that are made. My main concerns are:

**Authors (A):** We thanks the referee for his/her positive review. Please find our answers on referee's comments below.

**R:** (1) Was analysis done to calculate the correlation between the drivers? This was done between drivers and targets, but high correlations between drivers suggest the ML can do with utilizing fewer drivers. In the current state, there are too many drivers utilized for the findings to be of substantial use to the community without providing analysis that these drivers are available and colocated with the POC observations. Right now, the paper, with some minor revisions, would be useful to the BGC modeling community only.

**A:** The analysis of the correlation between drivers has been done but after consideration we prefer to not include this information in this manuscript but explore this aspect in more detail as part of a next step when we apply the ML method on observations. Indeed, as the referee points out, the number of drivers can be

minimised through the strong correlation between some of them. It also can be useful information in case when one of the drivers is not available in the real-world observation. The most problematic drivers from real-world data are Plankton Functional Types (PFTs) carbon concentration (as mentioned in the text, lines 506-507 in the revised manuscript). We found that there are correlations between some PFTs, and between PFTs and environmental conditions. However, the estimated correlation does not provide any specific information on the possibility for driver replacement, which would require further work. The intention of our work was to investigate the large spectre of drivers. Moreover, our work is based on the modelled data, and we used only one biogeochemical global ocean model PlankTOM (as mentioned in the text, lines 528-529 in the revised manuscript). Thus, any correlation between drivers can result from the physics as well as from the model construction. To avoid any biased conclusion and provide a large spectre of possible drivers and its role on the POC distribution we limited our analysis by the correlation between targets and drivers.

We added in the text (lines 291-293 in the revised manuscript): *Correlations between drivers could also provide valuable information to minimise the number of drivers but they are not shown here where the focus is on discovering the effect of a large set of drivers on POC distribution, and because driver correlations could also result from the physics as well as from the model construction.*

**R:** (2) Was analysis done to determine which driver observations we do have (and at which time/location)? Is there a set of drivers that can be tested that would correspond to what is currently available (and could be utilized in the near future within ML)? Right now, I don't forsee the current findings to be directly applicable, as most of the driver sets are large. Do all the observations of POC also have observations of all the drivers? What set of drivers would be realistically available for use in ML? It would be most useful to start there and then add individual drivers to see which additions have the largest impact on the ML results.

**A:** We did not explore specifically the availability of drivers' data in all stations where the measurements of POC were available, but rather tested the ML approach in ideal conditions of data availability. For example, measurements of PFTs carbon concentration are rare. Directly applying our method using only available observations would be difficult to justify without further testing due to the insufficient number of observations that probably would not capture the POC variability. Our study shows the validity of the ML method and provides information on the observations needed to reconstruct POC distribution in the ocean. A lot of work is ongoing to develop alternative estimates of a range of ecosystem-related data which will become available to use as drivers. Some of the modelled variables could also be used as drivers if they are sufficiently validated against observations before their use.

We added the following points in the discussion in the revised manuscript (lines 509-520): *Testing the present ML approach on observations will also help provide suggestions for an optimal set of drivers that can be measured specifically for POC reconstruction. For example, based on model results only, our results suggest that microzooplankton concentration is particularly important and should be measured more systematically, especially in the regions of high interannual variability. Likewise, this work provides information on the variables that are less important in POC variability, like vertical profiles of gelatinous zooplankton, or mixed phytoplankton for $POC_S$ and coccolithophore for $POC_L$, and, thus, less important to be measured in this context. These results will need to be tested with observations before firmly confirming the validity of the drivers. The validated driver sets can help guide observational programs. In addition, recent advances in plankton imaging (Irisson et al., 2022; Lombard et al., 2019; Orenstein et al., 2022) and omics (Faure et al., 2021) will soon provide a new global set of data to estimate PFT concentrations across ocean basins allowing to better identify potential biological drivers of POC variability. The new available data of PFTs will significantly facilitate the application of ML methods, such as the one developed here, to observational data.*

*Faure, E., Ayata, S.-D., & Bittner, L. (2021). Towards omics-based predictions of planktonic functional composition from environmental data. Nature Communications, 12(1), 4361. https://doi.org/10.1038/s41467-021-24547-1*

*Irisson, J.-O., Ayata, S.-D., Lindsay, D. J., Karp-Boss, L., & Stemmann, L. (2022). Machine Learning for the Study of Plankton and Marine Snow from Images. Annual Review of Marine Science, 14(1), 277–301. https://doi.org/10.1146/annurev-marine-041921-013023*

*Lombard, F., Boss, E., Waite, A. M., Vogt, M., Uitz, J., Stemmann, L., et al. (2019). Globally Consistent Quantitative Observations of Planktonic Ecosystems. Frontiers in Marine Science, 6, 196. https://doi.org/10.3389/fmars.2019.00196*

*Orenstein, E. C., Ayata, S., Maps, F., Becker, É. C., Benedetti, F., Biard, T., et al. (2022). Machine learning techniques to characterize functional traits of plankton from image data. Limnology and Oceanography, 67(8), 1647–1669. https://doi.org/10.1002/lno.12101*

**R:** (3) Whether with your best ML results, or with the most feasible set of observed drivers, I am curious as to why no analysis was done about where additional observations would be of most value to the ML - where added observations would alter the global performance of the ML?

**A:** This is an interesting question and results of such work can be useful for future deployment plans. However, it will represent another study dedicated to the Observation System Simulation Experiences which study the sensitivity of different regions to the number of available data. In this study we wanted to investigate what

available data of POC concentration and potential drivers (modelled or observed) can tell us about the linkage between surface ecosystems and sinking carbon. We extended the discussion to include some ideas and suggestions about the regions that need more observations based on the current analysis (lines 497-499 and 510-512 in the revised manuscript).

**R:** (4) Based on Figure 7, there are many features with extremely low importance. Did you go a step further and test your ML approaches and driver sets without these low-impact drivers?

**A:** We did test the effect of removal of low-impact drivers, but the improvement of accuracy was not significant. The results can be interpreted as some of the drivers do not play an important role in the POC concentration distribution, but there is no driver that particularly reduces the accuracy. From figures 5 and 6 we can see that for $POC_S$ reconstruction the accuracy is very sensitive to the presence of microzooplankton in driver set, and for $POC_L$ reconstruction the accuracy is sensitive to the general presence of PFTs and varies the small range. We added in the text (lines 362-363 in the revised manuscript): *It is worth noting that any driver that shows negative importance in the reconstruction has only a small influence on the accuracy (Figures 5 and 6). Thus, its removal does not improve the reconstruction significantly.* We added a sentence related to the less important drivers to our answer on referee point (2) as well.

**R:** In summary, as it currently stands, the article is most useful for BGC modelers. I think this will be a great contribution beyond this community, after care is taken to critically think (and analyze if any are redundant to the ML) which drivers are realistically available now at these observations, and if these prove to be inadequate for the ML results, determine the smallest set of additional environmental conditions that must be observed for the ML to give good results. If the current set of observed conditions are adequate, then many additional experiments can be done. Such as when and where do we need to sample to improve our ML model?

**A:** Thank you for this comment. As mentioned by the referee, we also believe that this work will contribute to further model development. Moreover, we believe that it will motivate other works, particularly in Observation System Simulation Experiences to identify key regions for future cruises.

The article suggests that some of the drivers are particularly important and some not. We added in the conclusion (lines 510-514 in the revised manuscript): *For example, based on model results only, our results suggest that microzooplankton concentration is particularly important and should be measured more systematically, especially in the regions of high interannual variability. Likewise, this work provides information on the variables that are less important in POC variability, like vertical profiles of gelatinous zooplankton, or mixed phytoplankton for $POC_S$ and coccolithophore for $POC_L$, and, thus, less important to be measured in this context.*

However, our results are based on the output of one biogeochemical model and, thus, can be biased (lines 528-531 in the revised manuscript). Further analysis based on observation is required to confirm our findings and guid observational programs (lines 514-515 in the revised manuscript). Moreover, the ongoing work on PFTs' measurements will facilitate this analysis and help to confirm our findings (lines 516-519 in revised manuscript).

---

## Author Comment (AC2)

Answer to Referee #2:

**Referee (R):** The manuscript was a joy to read. The authors evaluated the potential of using XGB and random forest to reconstruct POC in the global ocean. The authors found XGB performs marginally better than random forest.

The authors described the method very well and like that that testing was in regions away from training data.

**Authors (A):** We thank the referee for the positive feedback.

**R:** I would consider explaining variable importance and what it physically means. The authors note that the sine of latitude is in the top 10 drivers that influence large POC. Is this to be expected?

**A:** The role of latitude in the distribution of large POC was expected as there is a lot of meridional variability of POC distribution, which we can see in Figure 8b. We added in the text (lines 357-358 in the revised manuscript): *$POC_L$ distribution has a lot of meridional variability that results in the sinus of latitude being in the top 10 drivers.*

**R:** My only question is do ML algorithms like this one have the potential to replace parameterizations in ocean models?

**A:** Machine Learning techniques can help to improve parameterisations in ocean models. For example, ML can replace parameters used in parameterisation and make them region or/and time dependent as well as dependent on the ecosystem conditions as in our case with the POC concentration. We will not say that ML techniques will replace parameterisations, but they have the potential to improve it for sure. We added in the text (lines 521-526 in the revised manuscript): *The relationships between key variables and surrounding conditions based on Machine Learning can provide a new way for establishing parameters in ocean model parameterisation. The parameters can be time and space dependent and, thus, vary from one region to another better representing the physics. Relationship between POC concentration and environmental and ecosystem conditions can help to replace parameters in parameterised sinking velocity in PlankTOM. The reconstructed POC concentration over the global ocean will contribute to the reconstruction of porosity and opacity of particles that are key variables in the sinking matter velocity.*